# A strong statistical link between aerosol indirect effects and the self-similarity of rainfall distributions

Kalli Furtado[1] and Paul Field[1]

[1]Met Office, Exeter, UK

**Correspondence:** Kalli Furtado (kalli.furtado@metoffice.gov.uk)

**Abstract.** We use convective-scale simulations of monsoonal clouds to reveal a self-similar probability density function that underpins surface rainfall statistics. This density is independent of cloud-droplet number concentration and is unchanged by aerosol perturbations. It therefore represents an invariant property of our model with respect to cloud-aerosol interactions. For a given aerosol concentration, if the dependence of at least one moment of the rainfall distribution on cloud-droplet number is a known input parameter, then the self-similar density can be used to reconstruct the entire rainfall distribution to a useful degree of accuracy. In particular, we present both single-moment and double-moment reconstructions that are able to predict the responses of the rainfall distributions to changes in aerosol concentration. In doing so we show that the seemingly high-dimensional space of possible aerosol-induced rainfall-distribution transformations can be parametrized by a surprisingly small (at most three) independent "degrees of freedom": the self-similar density, and auxiliary information about two moments of the rainfall distribution. Comparisons to convection-permitting forecasts of mid-latitude weather and atmosphere-only global simulations show that the self-similar density is also independent of model physics and background meteorology. A theoretical explanation for this invariance is given, based on numerical results from a stochastic rainfall simulator. This suggests that, although aerosol-indirect effects on any specific hydro-meteorological system may be multifarious in terms of rainfall changes and physical mechanisms, there may, nevertheless, be a universal constraint on the number of independent degrees of freedom needed to represent the dependencies of rainfall on aerosols.

## 1 Introduction

The indirect effects of aerosols on precipitation influence the Earth's hydrological, energy and geochemical cycles on a range of timescales. Moreover, because of diversity in model representations of aerosol-cloud interactions they are a large source of uncertainty in weather and climate predictions. These uncertainties have their proximal origins in what is essentially an

engineering problem: different communities of developers have adopted discrepant approaches for encoding aerosol-indirect effects into their models. However, these discrepancies are in part rooted in a lack of scientific consensus as to the mechanisms by which aerosols affect clouds and precipitation. These discrepancies were starkly highlighted by Khain et al (2008) who reviewed a large number of earlier studies and classified them according to whether increasing the concentration of aerosols increased or decrease the surface precipitation. They concluded that the precipitation response depends on such a large range of factors, including the macro-scale cloud regime, cloud microphysics, and the thermal and dynamic conditions of the ambient atmosphere, that a general, system-independent answer to the question "*do aerosols increase or decrease precipitation?*" is not possible. Furthermore, Stevens and Feingold (2009) suggested that the answer to this question may often be "*neither*", because systems of clouds adjust to counteract the aerosol-induced changes in precipitation. This implies that although an individual cloud may have a large response to aerosols, changes in the amount of precipitation average over an area may be much smaller. This was illustrated by Seifert et al (2012) who showed that aerosols had negligible effects on precipitation over a range of regional numerical weather predictions.

In the absense of a general theory, much research has focused on elucidating the physical mechanisms that determine the precipitation response (or lack of) on a case-by-case basis. Khain et al (2008) suggested that this could be done by considering how, for each specific system, the various sources and sinks of condensed water respond to aerosol perturbations. If the precipitation rate adjusts over time to a slowly varying state in which sources and sinks of condensate approximately cancel out, then an aerosol change which increases the sources more than it increases the sinks will necessarily lead to an increase in the amount of precipitation. Therefore, if we consider a two systems both precipitating at rate, $P$, and subject one to an aerosol perturbation, the perturbed system will evolve to a new state with a (possibly) different precipitation rate, $P + \delta P$, where $\delta P$ is due to a change in the net source of condensate relative to the unperturbed system. If we denote the response of the rate of production of condensate to an aerosol perturbation by $\delta G$ and the response of the loss rate by $\delta L$, then response of the mean precipitation rate, $P$, can be written heuristically as

$$\delta P = \delta G - \delta L. \tag{1}$$

The mean precipitation, $P$, therefore increases or decreases according to whether $\delta G - \delta L$ is positive or negative. In general, $\delta G$ is due to changes in condensation and atmospheric dynamics (i.e., convergence), and $\delta L$ is due to changes in evaporation and dynamics. In general the responses of these factors are strongly coupled, and are highly system dependent, whence the diversity in aerosol indirect effects originates. As an example of this diversity, we may follow Khain et al (2008) in contrasting subtropical stratocumulus with tropical-deep convection over oceans. Stratocumulus clouds, when capped by drier air in the free troposphere, exist in an evaporation-dominated regime, where an increase in aerosol concentration increases drying of the boundary layer due to enhanced cloud-top mixing (Ackerman et al , 2004; Xue and Feingold , 2006). By contrast, because maritime deep convective clouds exist in humid environments their response is dominated by increased generation of conden-sate as cloud-droplets ascend. Khain's source-sink approach focuses on understanding mean-precipitation changes. In some situations, the sensitivities of other hydro-meteorological quantities are of equal importance. For example, the sensitivity of the rainfall frequency, occurrences of extremely heavy rainfall, or other characteristics of rainfall variability. In the most general

case, we are therefore interested in the response of the frequency distribution of surface rainfall rates, $f(p)$, as the concentration of aerosols changes.

Given sufficiently detailed observations or models of a cloud or system of clouds, the source-sink framework of Khain et al (2008) provides a detailed understanding of why a particular precipitation response was measured or simulated. To the best of our knowledge it cannot at present provide prior predictions of why a particular combination of $G$ and $L$ changes was the *necessary* response to an aerosol perturbation. In other words, except for cases of simplified theoretical models (Albrecht , 1989; Twomey , 1966), we can only explain why a particular response was observed, not why it had to occur in preference to

any other candidate for the response. Tao et al (2012) gives several examples which highlight this difficulty, and show how it arises from the multiplicity of different cloud-aerosol interaction mechanisms, or 'pathways', which can operate in the same system and compete to determine the overall precipitation response. For example, in deep convective clouds, because increasing aerosols reduces the size of the cloud droplets but increases the liquid water path, warm-rain generation is suppressed but cold-rain generation may be enhanced. Therefore, a theory for the precipitation response of deep convection requires explanations

for why a particular balance was struck between these competing effects, in a given situation. This balance involves adjustments in many microphysical processes, so the theory should explain why these processes responded as they did (and not in some other way). A detailed theory for an aerosol-indirect effect should therefore be able to predict the following: for a given cloud system, which processes will respond to an aerosol perturbation? How large will these responses be, relative to each other? How and why do these process-level responses determine the overall adjustment in sources and sinks of condensate?

In purely formal terms, such a theory would predict the $\delta G$ and $\delta L$ as functions of an arbitrary aerosol perturbation, $\delta N_{\mathrm{a}}$. To our knowledge, there is currently no theory of cloud-aerosol interactions capable of doing this. Hence we need to resort to numerical simulations to determine the relationship between $\delta G$, $\delta L$ and $\delta N_{\mathrm{a}}$ empirically. Given the wide variety of clouds, and multiplicity of cloud-aerosol interaction pathways available to these systems, and the potential sensitivity of the pathways to cloud regime and ambient environment, it is not surprising that the reviews by Khain et al (2008) and Tao et al (2012)

found that many such empirical theories exist in the literature.

In this paper, rather than investigating the mechanisms by which aerosols affect precipitation, we will address two simpler questions the answers to which provide insights into the *structure* that an eventual mechanistic theory should have. These questions are:

– what are the fundamental ingredients that a mechanistic theory needs to predict?

– are there any constraints on how these ingredients can be transformed due to an aerosol perturbation?

We will deliberately avoid the question of *why* rainfall changes in given way, for a given cloud regime. Rather we will take a set of simulated sensitivities of rainfall distributions to aerosol perturbations as given, and ask how much information is actually needed to describe ("parametrize") these sensitivities. As such, we are not concerned with predicting by how much rainfall increases or decreases in response to aerosol changes (this fundamental question is outside the scope of the framework that

we will propose), but rather with identifying a minimal set of information that is required to make such predictions. We will

primarily consider these questions in relation to the area- and time-averaged statistics of simulated rainfall over a large domain for a case of typical case of summer monsoon rainfall over East Asia.

The first of the above questions is related to the concept of 'degrees-of-freedom': how many independent variables are needed to describe the effect of an aerosol perturbation? For this to be non-trivial, we obviously need to seek as minimal a set of fundamental parameters as possible. For example, the physical mechanisms governing the dynamics of stratus clouds are different from those governing deep convective clouds, hence it may not seem surprising that they can respond differently to aerosols. But which properties of these systems are essential to their differing responses? What is the minimal set of properties that a mechanistic theory of cloud-aerosol interactions needs to predict in order to specify the aerosol-response of a cloud system? We will show that, even if an 'aerosol effect' is interpreted in the general sense of 'any aerosol-induced change in the rainfall rate distribution', a relatively small number of parameters is sufficient. The parameters that we identify are related to the dependencies of the first and second moments of the rainfall rate distribution on cloud-droplet number concentration (CDNC) and aerosol number concentration (AC); information that is readily available from simulations and satellite retrievals.

The second question is related to the concept of invariants: are there any properties which remain unchanged when a system is perturbed into a higher or lower aerosol state? If we restrict attention to large enough volumes, and short enough timescales, then the total mass of water (including accumulated surface precipitation) and the total energy are obvious atmospheric quantities that are approximately conserved when the aerosol amount changes. These quantities are however of limited use for aerosol-cloud interactions because we are interested in the partitioning of mass between the condensed and gaseous phases and the effect that this partitioning has on the precipitation rate. A slightly more useful invariant would be a statistical property of rainfall that did not change. A candidate would be the total rainfall accumulated over a large enough area and a long enough time. In situations were this variable is conserved, it does furnish a useful constraint: that the frequency with which rainfall occurs must be inversely proportional to the mean-rainfall rate, as aerosol varies (e.g., if the intensity of rainfall increases, the frequency of occurrence must decrease to conserve the amount of rainfall). However, there are many situations, not least when considering individual clouds or short-time scale responses, when accumulated rainfall is not conserved. In this paper will we show that the rainfall rate distributions simulated by our cloud-aerosol interacting mesoscale model have a common, underlying 'shape' that is independent of cloud-droplet number concentration (CDNC) and independent of aerosol concentration. This underlying distribution can be extracted by re-scaling simulated rainfall rate distributions into a dimensionless form. Self-similarity of this kind is common in systems where dynamical processes occurring on different scales are stretched (or sped-up) copies of each other. Examples from atmospheric science include aggregation of ice particles (Field et al , 2005) and boundary layer turbulence. Moreover, Field and Shutts (2009) and Lovejoy et al (2008) showed that observed rainfall also has a self-similar rate distribution.

The existence of an invariant rainfall distribution in our simulations implies that the cloud-aerosol interactions act by squashing or stretching this distribution. We will show that these stretches amount to transforming how frequently rainfall occurs and how much rainfall occurs for a given number of cloud droplets. It is therefore because an invariant distribution exists that a succinct specification of cloud-aerosol interactions in terms of the CDNC-dependencies of only two moments of the rainfall rate distribution is possible. In addition will we show that a single-moment, e.g., rainfall frequency, is sufficient if it is supplemented

by an empirically determined frequency-amount relationship which can be used to predict the other required moment. We will show that these frequency-amount relationships can be parametrized as power-laws with aerosol-dependent parameters. This implies that systems with the same power-law exponents and the same invariant distribution will have the same sensitivities to aerosols.

## 2   Model configuration and simulation set-ups

The simulations analysed were performed with a convection-permitting (0.03° horizontal-resolution) configuration of the Met Office Unified Model. A description of the model set-up can be found in Furtado et al (2018), together with a detailed description of the non-aerosol components of the microphysics scheme (see also Grosvenor et al (2017)). The simulated domain is situated over central and southern China ($17°–35°$N,$97°–126°$E). The simulated period is from 12 UTC 17 May to 12 UTC 22 May 2016, during which time a cyclonic vortex formed in the lee of Tibetan Plateau (105E,30N) and propagated south-east across China over a 48 hour period (Furtado et al , 2018, 2020). The simulations use a double-moment version of the Clouds and AeroSols Interaction Microphysics scheme (CASIM), in which five species of hydrometeor (cloud, rain, ice, snow and graupel) have prognostic masses and number concentrations. The aerosol concentrations are initialised with horizontally homogeneous but vertically varying values that are allowed to evolve via advection, turbulent mixing and two-way coupling between cloud microphysics and aerosols. The initial vertical profiles are retained along the lateral boundaries to maintain a source of aerosol that is constant in time. Lateral boundary conditions for non-aerosol prognostic fields are derived hourly from a global-model forecast with the Met Office Unified Model. The primary meteorological fields (including winds, temperature and moisture) are re-initialised on a 36-hour cycle from Met Office operational global-atmospheric analyses. Only hours 12 to 36 of each forecast are used in this study, to avoid a spin-up period at the beginning of each cycle. This re-initialisation does not apply to the aerosol fields, which are free-running for the entire period. Hence, for each forecast after the 12 UTC 17 May cycle, the initial aerosol field is taken directly from the proceeding cycle.

The method for coupling clouds and aerosols is described in Miltenberger et al (2018): interstitial aerosol particles are removed by activation of cloud-droplets; following activation, a prognostic variable for dissolved aerosol mass is co-advected with the hydrometeors so that it is transported conservatively through clouds. During evaporation of hydrometeors, the soluble mass is re-deposited into the air with a number concentration equal to the number of evaporated particles. Therefore, redeposited aerosols usually have a mean size exceeding that of the previously activated aerosols (because collision-coalescence gives rain drops that are fewer in number than the cloud droplets from which they develop). Hence, aerosols that were activated from the "accumulation" mode can be converted to coarser particles during evaporation. Ice nucleating aerosol particles are not simulated by prognostics, but CDNCs can modulate ice hydrometeor concentrations via a temperature-dependent parametrization of immersion freezing.

We will compare the results from three simulations with different initial- and boundary-aerosol concentrations. These will be called the high-, intermediate- and low-aerosol concentration experiments. Particularly in the figures, it will be convenient to label these as *aero+*, *aero○*, and *aero-*, respectively. In the atmospheric boundary layer, the order-of-magnitude mass and

number concentrations in the high-aerosol experiment are $10^{-8}$ kg kg$^{-1}$ and $10^{10}$ kg$^{-1}$, respectively. The typical values of the spherical-equivalent mean radius are 0.1 microns and 1 micron for accumulation- and coarse-mode aerosols, respectively. The vertical profile for the intermediate-concentration regime was identified by taking average aerosol concentrations, over the target domain, from a global-model simulation with a multi-species aerosols and atmospheric composition scheme. The initial profiles for the high concentration experiment are obtained from the intermediate-aerosol profiles by increasing mass and number of accumulation and coarse mode aerosols at each level by a factor of 10 (thereby leaving the aerosol-particle size unchanged). A factor of 10 reduction was applied to generate the initial profiles for the low aerosol experiment. Vertical profiles of initial aerosol in the experiments can be found in the supplementary material.

## 3   The effects of aerosol-number concentration on the simulated rainfall distributions

We focus firstly on the deep convective regime, which we identify as columns with relatively large fractions of low-, mid- and high-level cloud (defined as area fractions of low-, mid-level and high clouds greater than 0.1, 0.6, 0.8, respectively). These criteria pick out features, such as cloud bands around the cyclonic vortex, and convective clouds over high terrain, which have unbroken layers of condensate distributed throughout the depth of the troposphere. Such features are the main producers of heavy rainfall in the simulation domain (further information on their characteristics can be found in the supplementary material).

For this regime, the effects of aerosols on precipitation can be seen in Fig. 2a which shows the rainfall rate frequency distribution functions for the three aerosol-concentration experiments. The frequency decreases with increasing aerosol concentration for most rainfall rates. In particular, light and heavy rain are more frequent when there is more aerosol. The differences between the simulated distributions reflect the totality of aerosol effects on precipitation in our model, in the selected cloud regime. Statistically, these differences are a result of either changes in the frequency of occurrence of cloudy columns in the selected cloud-fraction regime, or changes in the rainfall rates produced in these columns.

The aerosol indirect effects that are included in our model are mediated via changes in cloud-droplet number concentration (CDNC). Hence, a natural first step for investigating the relationship between aerosol and rainfall rate is to decompose the rainfall rate distributions into components from model columns with different CDNCs. We will use the vertically averaged CDNC in each column to represent the droplet numbers. Mathematically, we may write:

$$f(p) = \sum_n f_n(p) \Delta n, \tag{2}$$

where $n$ ranges over a set of prescribed CDNC intervals with widths $\Delta n$, and $f_n(p)$ is the frequency density of rainfall rate per unit precipitation flux, per unit CDNC. We will call the component distributions, $f_n$, the 'conditionally sampled rainfall distributions' (or just CDNC-conditioned distributions, for brevity) to indicate that they are constructed by sub-sampling the total rainfall rate distribution, based on CDNC.

The different colored lines in Fig. 2b show the CDNC-conditioned distributions which contribute to the rainfall distributions shown in Fig. 2a. The set of five logarithmically spaced CDNC intervals shown in the figure's legend has been used for the

decomposition. For each color (CDNC interval), the three lines (solid, dashed and dot-dashed) correspond to the three aerosol-concentration experiments. We see that increasing aerosol concentration increases the frequencies of rainfall rates occurring at relatively high CDNCs (see, e.g., the purple and red lines). This is consistent with an increase in the prevalence of high CDNCs when the aerosol-number concentration increases. For smaller CDNCs (see, e.g., the orange and blue lines), heavy rainfall is suppressed by increasing the aerosol concentration. The suppression becomes stronger, and extends to smaller rainfall rates, as the CDNC is lowered. The differences in the statistical properties of rainfall occurring at different CDNC and for different aerosol concentrations can also be seen in the moments of the CDNC-conditioned distributions (Fig. 3). The $k$th moment of the $n$th conditional distribution is defined by:

$$M_k(n) = \int p^k f_n(p) dp, \tag{3}$$

(where, numerically, the integral can be approximated for any set of precipitation flux intervals). The zeroth moment, $M_0(n)$, for each CDNC interval, is related to the frequency of rainfall, $\mathcal{M}_0$, via

$$\mathcal{M}_0 = \sum_n M_0(n) \Delta n, \tag{4}$$

(Note that the occurrence of the CDNC-interval width, $\Delta n$, is due to the definition of the $f_n$s as frequency *densities*, i.e., frequencies per unit CDNC, in Eq. (2).) In general, the $k$th moment of the rainfall rate distribution is $\mathcal{M}_k(n) = \sum_n M_k(n) \Delta n$. In particular, $\mathcal{M}_1$ is the domain averaged rainfall amount, which is closely related to the rate of convergence of moisture into the domain. It is convenient for the following to note that $\Delta \mathcal{M}_k(n) = M_k(n) \Delta n$, is the contribution to the total moment $\mathcal{M}_k$ that comes from the sub-sample of rainfall rates that occur with a CDNC of $n$.

Because they contribute to the rainfall frequency and rainfall amount, the CDNC-conditioned rainfall frequencies, $\Delta \mathcal{M}_0(n)$, and rainfall amounts, $\Delta \mathcal{M}_1(n)$, (and hence also their equivalent densities, $M_0(n)$ and $M_1(n)$) are of particular interest. Figure 3(a,b) shows how $M_0$ and $M_1$ vary with CDNC for each simulation. For high CDNCs, increasing the aerosol concentration increases the frequency and amount of rainfall. When the CDNC is low, aerosol increases have the opposite effect and suppress rainfall. Another defining characteristic of rainfall is the mean-rainfall rate (sometimes referred to as rainfall 'intensity'), which measures the average flux of rain at surface points where rain is falling, neglecting points with no or negligible rainfall. For each value, $n$, of the CDNC, we can define a CDNC-conditioned mean rainfall rate, $\lambda_n$, by the ratio

$$\lambda_n = M_1(n)/M_0(n). \tag{5}$$

This is the mean rainfall rate for the sub-sample of columns where rainfall reaches the surface and the column-averaged CDNC is $n$ kg$^{-1}$. The CDNC-conditioned mean rain rates are plotted in Fig. 3c, which shows that rainfall intensity increases with CDNC for large aerosol concentrations, but decreases with CDNC in cleaner conditions.

## 4   Self-similarity of simulated rainfall statistics

A noticeable feature of CDNC-conditioned distributions shown in Fig. 2 is the very large (six orders of magnitude) spread in rainfall frequency density as the CDNC varies. The inter-simulation spread for a fixed CDNC is also large: up to four orders of

magnitude, for some rainfall rates. However, there is some visual indication that the 'shapes' of the conditioned distributions are quite similar to each other. This suggests that a suitable simultaneous scaling of precipitation flux and frequency density might reveal that these distributions are re-scaled instances of a single, underlying frequency distribution. On dimensional grounds, we expect such a scaling to map rainfall rate to a dimensionless precipitation flux, and map the frequency densities, $f_n$, to dimensionless distribution functions.

An obvious candidate for re-scaling the precipitation flux for each CDNC is the mean-rainfall rate, $\lambda_n$, because it is the only combination of $M_1(n)$ and $M_0(n)$ that has the dimensions of a mass flux. Similarly, the quantity

$$\nu(n) = \frac{M_0^2(n)}{M_1(n)} \tag{6}$$

(plotted in Fig. 3d), is the unique combination of the zeroth and first moments that has the same dimensions as the frequency densities. For each aerosol concentration and CDNC, we will use these parameters to define a dimensionless rainfall rate, $r_n = p/\lambda_n$, and a dimensionless frequency density, $\Phi$, as follows:

$$f_n(p) = \frac{M_0^2(n)}{M_1(n)} \Phi\left(\frac{p}{\lambda_n}\right). \tag{7}$$

The colored lines in figure 4 show the dimensionless distributions, derived for each CDNC, $n$, and each aerosol concentration, as functions of their corresponding dimensionless rainfall rates, $r_n$. It is clear that this scaling results in a significant amount of data collapse: the disparate distributions shown in Fig. 2b give rise to very similar distributions in the re-scaled variables. The degree of similarity between the scaled distributions is sufficient for us to regard the ensemble mean of the scaled distributions as defining a single ('universal') dimensionless density function that is independent of both aerosol concentration and CDNC. The histogram of this universal distribution is shown by the black line in Fig. 4. We have tried unsuccessfully to fit a functional form to the universal histogram. This is not to say that no such form exists, but in the absence of one, a pragmatic approach is to use the histogram itself to *define* the universal distribution. We therefore view the distribution as the probability density of a random variable defined by randomly sampling the empirical histogram shown in Fig. 4. The data specifying the histogram is available in the supplementary data.

## 5 Reconstructing the rainfall distributions

If the proposed universal distribution is independent of CDNC and aerosol concentration, then we expect to be able to use the scale transformations in Eq. (7) to approximately reconstruct the rainfall rate distributions. In this section we assess the accuracy of these reconstructions. The simplest case occurs if both $M_0(n)$ and $M_1(n)$ are known functions of the CNDC, $n$. In this case, Eq. (7) can be used directly to estimate $f_n$ for each $n$, and the total rainfall rate distribution, $f$, can be estimated as the sum of the $f_n$s using Eq. (2). We will call this case a double-moment closure, because two moments of the CDNC-conditioned rainfall distribution are required. If only one moment of the first two moments is a known function of $n$, then additional information is required that parametrizes the other moment (or more generally the aerosol conditioned mean rain rate, $\lambda_n$, and normalisation, $\nu_n$) in terms of the known moment. Here will we consider the case where the CDNC-conditioned

frequency $M_0$ is the known moment, and derive an empirical closure relation that specifies $M_1$ in terms of $M_0$. This case will be called a single-moment closure, because only one moment of the CDNC-conditioned distributions needs to be specified. The double-moment closure is essentially just a further test of the validity of the data-collapse affected by the rescaling the rainfall rate distributions. However, we show in Section 5.1 that it permits an insight into aerosol indirect effects because it separates the contributions of changes in the rainfall intensities, $\lambda_n$, from changes in the frequencies of occurrence rainfall (as represented by $\nu_n$). The single-moment closure (Section 5.2) is a more stringent test of the universality of the invariant distribution; moreover, it allows us to identify the minimal information that is needed to parametrize the effects of aerosols on rainfall rate distributions.

## 5.1 Double-moment closure

If the first two moments of the CDNC-conditioned frequency distributions, $f_n$, are known then the total rainfall distribution for each simulation can be reconstructed using Eqs (2) and (7). The black circles in Fig. 5a show this reconstruction for the high-concentration simulation. The solid black line shows the rainfall rate distribution obtained from the simulation. The total frequency distribution is the sum of CDNC-conditioned contributions, $f_n \Delta n$, from each of the specified CDNC intervals. These contributions are shown by the colored lines in Fig. 5a. The estimated CDNC-conditioned densities are shown by the colored circles. In general, the double-moment reconstructions can reproduce the conditioned distributions and the rainfall rate frequency distribution. The accuracies of the reconstructions for the other two aerosol-concentration experiments are similar.

Overall, the reconstructions are accurate enough to predict the effects of aerosols on the rainfall distributions. These effects can be seen in Figs 5(b,c), which show the fractional changes in the rainfall rate distributions, compared to the high-concentration simulation. The black-dashed lines and symbols show the fractional changes in the simulated and predicted total rainfall rate frequency distribution. The dashed color lines and symbols show the fractional changes in each CDNC-conditioned distribution. The double-moment reconstructions are able to capture the changes in rainfall rate frequencies due to the aerosol perturbations. Moreover, the relative contributions to the overall changes coming from each CDNC interval are also predicted well. For example, the reductions in the frequencies of heavy rainfall as aerosol concentration decreases are predicted, and the relative importance of high CDNCs for driving these reductions (red lines and symbols) is also captured.

The double-moment reconstructions allow the contributions of the CDNC-conditioned mean rainfall rate changes to be separated from changes in the CDNC-conditioned rainfall frequencies. These contributions can be inferred as follows. Firstly, we denote the high-aerosol concentration experiment as

$$f_{\mathrm{ref}}(p) = \sum_n \nu_{\mathrm{ref}}(n) \Phi\left(p / \lambda_{\mathrm{ref}}(n)\right) \Delta n. \tag{8}$$

For one of the other aerosol experiments, we may estimate the rainfall rate distribution after a reduction in aerosol concentration, based on the assumption that the conditional intensities, $\lambda_n$, do not change from their reference values, $\lambda_{\mathrm{ref}}(n)$. This assumption corresponds to the simplification that aerosol perturbations can alter the number of cloud droplets, but that this does not affect the intensity of rainfall for a given CDNC. Making use of the universal distribution, $\Phi$, this estimate is given

by:

$$\tilde{f}_{\exp}(p) = \sum_n \frac{\nu_{\exp}(n)}{\nu_{\mathrm{ref}}(n)} f_{\mathrm{ref}}(p, n)\Delta n. \tag{9}$$

Note that only the normalised frequencies, $\nu(n)$, have changed from their reference values, whilst the intensities, $\lambda(n)$, remain the same as in the high-aerosol experiment. Equation (9) says that if an aerosol perturbation does not affect the intensity of rainfall for a fixed CDNC, then the aerosol effect amounts to a rainfall-rate independent re-weighting of the relative contributions from each CDNC-conditioned distribution. For a given value of $n$, the re-weighting factor, $\alpha_n = \nu_{\exp}(n)/\nu_{\mathrm{ref}}(n)$ will enhance ($\alpha_n > 1$) or suppress ($\alpha_n < 1$) rainfall frequency, for that number of CDNC, uniformly across the rainfall rate spectra. The estimated fractional changes, assuming no changes in the CDNC-conditioned rainfall intensities, are shown by the solid black lines in Figs 5(b,c). By comparing these to the dashed black lines, it can be seen that they significantly overestimate the suppression of rainfall frequency with decreasing aerosol concentration, and do not capture the dependence of the fractional changes on the rainfall rate, $p$. For each CDNC, $n$, the short colored horizontal lines show frequency suppression factors, $\alpha_n$. For CDNCs such that $\alpha_n < 1$, the effects of aerosol on normalised rainfall frequency is such that rainfall is suppressed. Where the fractional change in a CDNC-conditioned distribution (the dashed-colored lines) is greater than the corresponding value of $\alpha_n$, the mean rainfall rate, $\lambda_n$, for this CDNC has increased (i.e., the rainfall produced for this CDNC is becoming more intense in response to decreasing aerosol). The inability of the constant-intensity estimates to predict the simulation results implies that the suppression of rainfall frequency by decreasing aerosol is therefore partly offset by the simultaneous intensification of rainfall intensities for some value of CDNC.

## 5.2 Single-moment closure

The double-moment closure described in Section 5.1 provides a test of validity of the universal distribution. It also allows us to separate aerosol-induced changes in rainfall intensity (at a fixed CDNC), from changes in the relative frequencies of rainfall occurring at different CDNCs. We will now investigate whether the effects of aerosols can be predicted using information about fewer than two moments of the CDNC-conditioned distributions. In particular, we will show that the CNDC-conditioned rainfall frequency, $M_0$, is sufficient, if it supplemented by an empirical closure-relation for $M_1$.

We have found that $M_0$ and $M_1$ can be related to each other by

$$M_1 = x M_0^y \tag{10}$$

where the parameters $x$ and $y$ vary between the experiments (see supplementary Fig. S4(a)). Furthermore, we found that the parameters $x$ and $y$ are not independent, and are related by $\log(x) = n_0 + ay$ where $n_0$ and $a$ are constants that are independent of the aerosol-concentration, implying that a single, aerosol-dependent parameter (either $x$ or $y$) is sufficient to specify the relationship between $M_0$ and $M_1$ (Fig. S4(b)). Equivalently, it is convenient to express both $y$ and $x$ parametrically as functions of the ratio, $N_{\mathrm{a}}/N_{\mathrm{ref}}$, of the initial aerosol concentration in each experiment to the concentration, $N_{\mathrm{a}}$, in the intermediate-aerosol concentration experiment:

$$y = \delta + \gamma \log(N_{\mathrm{a}}/N_{\mathrm{ref}}), \qquad x = \epsilon + \eta \log(N_{\mathrm{a}}/N_{\mathrm{ref}}) \tag{11}$$

The parameters $\delta, \gamma, \epsilon$ and $\eta$ are given in Table 1. Figure 6a shows that there is good agreement between simulated first moments, $M_1(n)$, and the prediction obtained from the empirical fits.

The empirical relationship between moments can be used to replace $M_1(n)$ in Eq. (7) by a function of $M_0$. This gives
the rainfall frequencies distributions as functions of $M_0$ only. The utility of the single-moment closure is assessed in Fig. 6b, which compares the simulated rainfall amounts, $\mathcal{M}_1^{>p_0} = \int_{p_0} f(p)dp$, above each of a range of rainfall rate thresholds, $p_0$, to the predictions obtained from the single-moment closure. The good agreement obtained indicates that the simulated aerosol indirect effects can be fully parametrized by the CDNC-dependence of the rainfall frequency.

We will summarise the overall indirect effects of aerosols on precipitation by the first four moments, $\mathcal{M}_0, \ldots, \mathcal{M}_3$, of the total
rainfall rate frequency distributions in each of the simulations. The black lines in Figure 7 show these moments as functions of the initial aerosol concentrations (where the latter are expressed relative to the intermediate-aerosol experiment). The variations of these moments express different aspects of the hydrological sensitivity of the system to perturbing the aerosols: the changes in the zeroth, and first moments correspond to the changes in the frequency of occurrence and amount of rainfall, respectively; the second and third moments, $\mathcal{M}_2$ and $\mathcal{M}_3$, express changes in the width of the rainfall distributions, particularly the relative
frequencies of occurrence of large rainfall rate. The symbols in Fig. 7 show the values of these moments predicted by the double-moment (blue) and single-moment (black) closure relations. In most cases, the predictions are able to reproduce the simulated values of the moments reasonably well. Moreover, the predictions capture the increasing trends in the moments as the aerosol concentration increase The agreement is slightly less good for some values of the single-moment reconstructions and for the highest-order moment tested. This indicates that the predictions are accurate enough to reproduce the sensitivity for
the simulated rainfall to aerosol perturbations. The double-moment closure is typically more accurate than the single-moment closure, as expected because it contains more information about the CDNC-dependence of rainfall statistics.

## 6   Regime dependencies

So far we have considered a deep convective regime, where there are relatively large cloud-area fractions at low-, mid- and high-levels. Specifically we selected only model columns where the low-, mid- and high-cloud area fractions, exceeded 10,
60 and 80 percent, respectively. This is a computationally simple way of selecting deep-convective columns, such as those associated with the eastward propagating vortex (see Figs S1 and S2 in the supplementary material). There are, however, other regimes of clouds and precipitation occurring within the domain during simulations, which may respond to aerosols differently from the deep-convective regime.

Regimes for which we may expect aerosol-cloud interactions to differ from those in deep convection are stratus clouds and
shallow convection. In general, we may expect different responses to aerosols in regimes where mid- and high-level clouds are present because this may be related to the role of ice-phase processes. For example, Fig. 1 and supplementary figures S1 and S2, show that there is a region of precipitating low-level clouds in the wake of the cyclonic vortex. These clouds are readily identified as emitting higher fluxes of long-wave radiation than the deep convective-cloud regime (Figs 1c,S2), suggesting that they exist in a regime dominated by warm-cloud microphysical processes. In this section, we will extend the proceeding

analysis to a range of cloud-fraction regimes, chosen to span the cloud-types present in the simulations. We will assess the extent to which aerosols affect these regimes differently, and the extent to which they exhibit the self-similarity in rainfall statistics identified above for the deep-convective regime.

We will denote the low-, mid- and high-level cloud fractions by $L, M$ and $H$, these are defined from the model's sub-grid cloud-area fractions using the ISCCP cloud-height pressure classification. We divided the model output into three mutually exclusive categories: the deep convective cloud regime, with high fractional cloudiness, described above for which $L > 0.1, M > 0.6$ and $H > 0.8$; a regime dominated by low-clouds, where $L > 0.9$, $M < 0.2$ and $H < 0.2$; a transitional/"marginal" regime with intermediate values of mid-level and high clouds ($L > 0.4, 0.4 < M < 0.6$, and $0.2 < H < 0.8$). In the supplementary material, we show that these categories divide the rainfall frequency distribution into a heavily precipitating (and highly cloudy) regime, a moderately precipitating regime with intermediate cloudiness, and less-cloudy and more lightly precipitating regime (supplementary Fig. S3). We also consider the totality of all cloudy columns, for which at least one $L, M$ or $H$ is non zero.

Figure 8a shows rainfall rate distributions for each of the selected regimes. We see that the deep convective regime (black) accounts for the majority of the rainfall occurring in the simulations. The other two regimes produce progressively less precipitation as the amount of high- and mid-level cloud decreases. For each regime, the CDNC-conditioned rainfall rate distributions are calculated then rescaled to their dimensionless forms using their corresponding mean-rainfall rates and normalised frequencies. The re-scaled distributions for each regime are plotted in Fig. 8b, from which it can be seen that the universal distribution is highly consistent across the regimes (except for the largest values of the dimensionless rainfall flux, where the universal histograms become regime dependent, perhaps because these values are relatively under-sampled for the moderate and lightly precipitating regimes). However the empirical relationships between the CDNC-conditioned zeroth and second moments *are* regime dependent (supplementary Fig. S4). Hence the parameters in the single-moment closure vary across the cloud regimes (Table 1). This is to be expected because the relationship between rainfall frequency and rainfall amount depends on the specific cloud-microphysical and macrophysical processes leading to rainfall, whereas we claim that universal distribution does not.

For each cloud regime, we can define the overall sensitivity of rainfall to aerosols by the differences in the moments of the rainfall frequency distribution between the high and low-aerosol concentration experiments, e.g., for the $k$th moment we have

$$\gamma_k := \frac{\Delta \log \mathcal{M}_k}{\Delta \log(N_a/N_{\mathrm{ref}})}, \tag{12}$$

where a $\Delta$ denotes the difference in its antecedent quantity between the two experiments. For moments $\mathcal{M}_0, \ldots, \mathcal{M}_3$, these quantities describe the sensitivities for rainfall frequency, rainfall amount and rainfall variability to aerosol perturbations. The sensitivities, $\gamma_0, \ldots, \gamma_3$, for each regime are plotted in Fig. 9 for the simulations (lines) and single- and double-moment closures (symbols). It can be seen that the predictions are in quantitative agreement with the simulation results in the three regimes, and for the totality of precipitating, cloudy columns. We see that the deep-convective cloud regime has a positive aerosol indirect effect for all four moments ($\gamma_k > 0$, $k = 0, \ldots, 3$). This is because increasing the aerosol concentration increases the rainfall frequency, amount (Fig. 7(a,b)) and the occurrence of heavy rainfall (Figs 7(c,d), 2a) in this regime. By contrast the low-cloud dominated and intermediate regimes have negative sensitivity, particularly for rainfall frequency and amount, because

increasing aerosols reduces rainfall in these regimes (Fig. 8a). Interestingly, when the domain is considered as a whole, the overall sensitivity of the entire system to aerosol perturbations is small. This is because of the opposing signs of the aerosol effects in different parts of the domain.

## 7   Discussion

Cloud-aerosol interacting systems show a range of responses to aerosol perturbations (Khain et al , 2008; Tao et al , 2012), from precipitation suppression (in, e.g., stratocumulus (Xue and Feingold , 2006; Ackerman et al , 2004), shallow cumulus and some deep continental clouds (Khain et al , 2008)) to precipitation enhancement in deep convection over oceans and, for some cases, deep convection over land. Ultimately, the change in the frequency distribution of rainfall rates induced by a change aerosol is a function of how all the hydro-meteorological processes occurring within the system respond to the aerosols. This includes modifications of the rates of condensation, $C$, and evaporation, $E$, and adjustments in the dynamical state, $D$, of the system. The aerosol-induced change, $\delta f$, in the rainfall rate distribution is given by a generalisation of Khain's 'source-sink' framework that also includes dynamical factors:

$$\delta f(p) = \delta\left(C - E - D\right)\frac{df}{dp} \tag{13}$$

Unfortunately, since $\delta C$, $\delta E$ and $\delta D$ are complicated combinations of many variables (Khain et al , 2008) and therefore $\delta(C - E - D)$ is not a simple function of $p$, Eq. (13) cannot be used directly, and we typically resort to numerical experiments to determine the factors influencing $\delta f$ in each particular case. The aim of such analyses is often to investigate how individual process rates have responded to an aerosol perturbation and to understand the effects that these have on precipitation. However, to our knowledge, several more basic questions are overlooked by this procedure. Firstly, how many parameters are actually needed to specify $\delta f$? (The variety of possible responses suggest that this parameter space is a high-dimensional one.) Secondly, can $\delta f$ be an arbitrarily complex perturbation, or are there any constraints on how cloud processes can adjust to redistribute rainfall over a range of intensities? In particular, are there any properties of rainfall rate distributions that we can expect to be *unchanged* by aerosol perturbations?

### 7.1   Is there a statistical property of rainfall that is invariant under aerosol perturbations?

We showed in Section 4 that, for any CDNC, $n$, and aerosol concentration, $N_{\mathrm{a}}$, the probability distribution defined by

$$\Phi(r) = \frac{1}{\nu_n} f_n\left(\lambda_n r\right), \tag{14}$$

where $\nu_n = M_0^2(n)/M_1(n)$ and $\lambda_n = M_1(n)/M_0(n)$, is independent of $n$ and $N_{\mathrm{a}}$. In Section 6 we showed that this distribution is also quasi-independent of the cloud regimes simulated. For example, it is approximately the same in low-cloud dominated regions and in regions of deep convection (except for large and small $r$). We do not know if the distribution is independent of the modeling system used, but this will be interesting to investigate in future work. Similarly, our model does not include prognostic ice nucleating aerosols, it is possible that these might alter the universal shape. The existence of this distribution implies

that rainfall events occurring with different CDNCs are statistically similar, in the mathematical sense that their frequency distributions can be transformed to each other by a change of scales. Equivalently, suppose we have a pair of equally long time series of precipitation values from two rainfall events, one with CDNC, $n_1$, another with CDNC, $n_2$. We can view both these time series as realisations of two different random variables, $p_{n_1}$ and $p_{n_2}$. The invariant distribution implies that the random variables $r_1 = p_{n_1}/\lambda(n_1)$ and $r_2 = p_{n_2}/\lambda(n_2)$ are identically distributed (with distribution $\Phi$) and independent of the value of $n$.

## 7.2 How many degrees of freedom are needed to describe aerosol indirect effects on precipitation?

The single-moment and double-moment reconstructions in Section 5 show that precipitation state of a cloud-aerosol-interacting system is specified by:

1. the universal distribution, $\Phi$, for the system, which is independent of the CDNC and aerosol concentrations in the system,

2. either

    (a) the zeroth moment, $M_0(n)$, and first moment, $M_1(n)$ of the CDNC-conditioned rainfall rate distributions, $f(p,n)$, as functions of the CDNC, $n$, or

    (b) a single moment (e.g., $M_0(n)$), and an aerosol-dependent, frequency-amount relationship, e.g., $M_1(n) = I_{N_a}[M_0(n)]$, for the system, which diagnoses the remaining moment (e.g., $M_1(n)$) in terms of the known moment,

From this information, the rainfall rate distributions of the system can be reconstructed to the degrees of accuracy demonstrated in Section 5. The choice of $M_0$ and $M_1$ is arbitrary: as shown by Field and Shutts (2009), any pair of moments could be used for the reconstructions.

For our simulations, the moment relation, $I_{N_a}$, can be parameterized as a family of power-law relationships of the form

$$M_1(n) = x(y)M_0(n)^{y(N_a)}, \tag{15}$$

between the CDNC-conditioned moments, where only one of the parameters, $x$ or $y$, needs to be specified directly in terms of the aerosol concentration. The remaining parameter (in this case, $x$) is a function of the other. We found that a further pair of power laws, $x \sim y^a$ and $y \sim N_a^\gamma$, were sufficient to specify the parameters in the moment relations. Hence a total of four constants (see Table 1) is needed to specify the mapping from the universal distribution to the dimensional rainfall distributions.

## 7.3 How "universal" is the invariant distribution?

Based on a single set of simulations it is not possible to evaluate the "universality" of $\Phi$ distribution. We have shown that the distribution is approximately independent of CDNC and cloud-regime, but dependencies on the modeling system, parameterization schemes and background meteorology have not been explored. In this section we present evidence which suggests that the distribution $\Phi$ is a statistically robust feature of global and regional simulations with the Met Office Unified Model. Firstly, we show that rainfall rate frequency distributions from twenty-three case studies of mid-latitude weather systems forecast with

a regional model configuration over the United Kingdom (UK) have the same $\Phi$ distribution as the (subtropical) May 2016 case
study. Secondly we show that the frequency distributions of daily mean rainfall over three $17° \times 12°$ regions from a 20-year
global climate simulation also give the same $\Phi$ as the regional models. In addition we show that in the global simulation the
rainfall rates from the model's microphysics and convection schemes both rescale to the same dimensionless distribution.

The regional model used for the UK case studies has the same physical parametrizations as the configuration described in
Section 2, except for the representation of aerosols. In the UK forecasts, aerosols are modelled with a single mass prognostic
(representing the combined mass of aerosols) which is produced from surface sources and advected. An aerosol number con-
centration is diagnosed as described by Wilkinson et al (2012) and then passed to the microphysics scheme which calculates
activation increments to the CDNCs. The same cloud microphysics scheme (CASIM) is used for both the UK and China cases.

The global model configuration is a version of the Met Office Global Atmosphere (GA; Walters et al (2019)). This config-
uration uses a single-moment microhysics scheme (Wilson and Ballard , 1999), coupled to the UK Chemistry and Aerosols
(UKCA) model which provides diagnosed CDNCs to the microphysics schemes. Hence the clouds and aerosols microphysics
in the global model is structurally different from that used in the regional models. Moreover, the global configuration includes
parametrized, sub-grid scale convection, which is an additional source of diversity between models.

For the UK case studies, we partition the rainfall into CNDC intervals, as described in Section 4, and cloud-fraction regimes.
We choose the cloud-fraction partition to reflect the most common local precipitation regimes, which are (very broadly): frontal
systems (with large amounts of high-clouds); stratiform cloud decks or shallow convection (with low-level cloudiness, but little
high cloud); an 'intermediate' regime, encompassing other combinations. The results are not sensitive to this classification.
Figure 10a shows that the rainfall rate distributions are less sensitive to CDNC than in the China case. The sensitivity of rainfall
to CDNC is muted because the simplified aerosol physics in the UK configuration produces less variable aerosol concentrations.
However, the combined variability across all CDNC intervals and cloud regimes is still many orders of magnitude. Figure 10b
shows that non-dimensionalization collapses the frequency distributions to single distribution. Moreover, this distribution is
the same function, $\Phi$, of non-dimensional rainfall, that was found in the China case study (the latter is reproduced in Fig. 10b
as the red histogram).

The UK and China forecasts both use the CASIM microphysics scheme. To assess the effects of differences in cloud mi-
crophysics, we calculate the non-dimensional distributions of daily-mean rainfall rates from a 20 year global, atmosphere-only
simulation using the Atmospheric Model Incomparision Project (AMIP) protocol. Rainfall rates during June, July and August
are selected for three geographical regions: the north-eastern Pacific ($20°$–$37°$N, $220°$–$245°$E); the western tropical Pacific
($2°$–$19°$N, $132°$–$157°$E); the Southern Ocean ($52°$–$70°$S, $206°$–$231°$E). This samples subtropical stratocumulus, tropical deep
convection and mid-latitude cyclones; thus each region tests for occurrence of universality in a different background climate.
Because a daily CDNC diagnostic is not available in the output from the simulation used, we partition the rainfall distributions
in each region into intervals of the cloud albedo calculated by radiation scheme. Although not equivalent to the CDNC parti-
tioning used for the regional simulations, this provides another way of classifying the rainfall with a property of clouds that
is sensitive to aerosol-cloud interactions. Figure 11(a,b) shows the frequency distributions of rainfall from large-scale clouds
(i.e., the microphysics scheme) and convective clouds (the sub-grid convection scheme), in each albedo interval (shading), for

each region (color). As expected, relatively heavier precipitation is associated with higher albedo in all regions and the balance between large-scale and convective precipitation varies between the regions. The large diversity of rainfall distributions is removed by non-dimensionalization, which approximately collapses all the frequency distributions onto the same dimensionless distribution. In particular, distributions from different albedo intervals and in different regions become very similar after rescaling. For convective and large-scale rainfall we define a universal distribution by the means of the corresponding dimensionless distributions. These are shown by the dashed and dot-dashed histograms in Figure 11c. We see that, despite structurally different treatments of precipitation processes in large-scale microphysics and sub-grid-scale convection, the non-dimensional histograms for the two schemes are approximately the same. Moreover, these distributions are also very similar to the universal distribution obtained from the convection-permitting forecasts over China (the solid black histogram). The degree of data-collapse (across regions, models, and cloud properties) suggests that the distribution $\Phi$ may be considered a 'universal' property of the Met Office model.

## 7.4 An explanation for the simulated rainfall universality

The existence of a widely applicable non-dimensional rainfall distribution suggests that it is due to underlying physical characteristics of rainfall producing processes that are independent of cloud properties, cloud type, meteorological conditions and model parametrizations. These factors determine the dimensional rainfall frequency distribution at a particular location and time, but their effects can be described by a two-parameter rescaling of the universal, underlying $\Phi$ distribution. In this section we will support this claim by showing that a simplified, 'toy' model, based on modeling rainfall as multi-scale stochastic process, can produce synthetic rainfall statistics which also have have a two-parameter family of distributions. The simplified model is based on the stochastic rainfall generator analysed by Rodriguez-Iturbe et al (1984). It simulates a discrete rainfall time-series with integer length $T$ as a sum of independent, temporally overlapping rainfall 'events'. Each event, $e$, is described by a duration, $d_e$, and intensity, $\lambda_e$. At each time step, $k = 1, \ldots, T$, a new rainfall event is initialised with a fixed probability $q_i$. Hence the total rainfall, $\hat{p}(k)$ is the sum of $\lambda_e$ for all events whose durations span across the time $k$. This generates rainfall time series that are sums of rectangular pulses with random lengths and heights. Models of this type are known to exhibit a rich statistical behaviour and have been used to generate synthetic rainfall rates for hydrological applications (e.g., Burton et al (2008)).

We make two further modelling assumptions: the event durations are numbers of time steps chosen independently from a power-law distribution (hence, $\Pr[d_e = t] \sim t^\alpha$, where $\alpha < 0$); the event intensities are uniformly distributed in an interval $[1, l_1]$. The parameters in the model are therefore the mean event intensity ($\overline{\lambda_e} = (l_1 - 1)/2$), the power-law exponent for the events durations ($\alpha$), and the initiation probability ($q_i$). These parameters can be varied to imitate external factors (e.g., aerosol perturbations) that affect rainfall event characteristics. For example, increasing the mean intensity (or, equivalently, the maximum rainfall intensity parameter, $l_1$) corresponds to a factor or process that increases the probability of heavier precipitation events and reduces the probability of lighter rainfall events. This is similar to increasing CDNC in our regional model simulations.

In Fig. 12 we show that the stochastic rainfall process can generate rainfall distributions with a self-similarity property that is reminiscent of Unified Model simulations. We do not attempt a detailed parametric description of the model's behaviour, because it suffices to show that there exist parameter regimes with universal distributions. Figure 12a shows examples of rainfall time series for a selection of values of of $\overline{\lambda_e}$ and $\alpha$ (with $q_i = 0.05$). As the mean intensity increases, the peak values of the precipitation time series increase; as the duration exponent increases, the rainfall rates become more correlated in time. Figure 12b shows the frequency distributions for several parameter combinations. Increases in heavy rainfall result from more intense rainfall events, or from longer duration events. The differences between the distributions are qualitatively similar to the effects of aerosols on the Unified Model simulations (e.g., Fig. 2). After renormalization these distributions become statistically near-identical (Fig. 12c), suggesting the existence of an exact self-similarity of rainfall statistics in this part of the parameter space. This similarity can be presumed to be a geometric property of the time series generated by the model.

The stochastic model encodes some basic properties of precipitation physics. Namely that rainfall at a location is a super-position of independent precipitation events with random durations and intensities. External factors, such as aerosols, alter the statistical distributions of the properties of rainfall events (e.g., increasing aerosols may make rainfall events heavier, on average). Hence, by analogy with the stochastic model, we propose that the universality of rainfall statistics in our regional and global simulations is consistent with the following ingredients:

1. rainfall statistics are due to the accumulation of quasi-independent rainfall 'events' with random durations and intensities;

2. external factors (e.g., aerosols) and cloud characteristics (e.g., stratiform or convective) affect rainfall-event properties, but as these properties vary the rainfall distributions remain within a family of self-similar distributions

We defer further investigation of these claims to future works.

## 8  Conclusions

There is no general theory for how aerosols affect precipitation. Instead, analyses of different cloud regimes have revealed a range of behaviours. Hence, at a theoretical level, a detailed understanding of how aerosols affect a particular cloud regime, or type of cloud-system, is probably the most that can be achieved. Such theories usually rely on numerical models to understand the physical mechanisms, or pathways, by which aerosols modify precipitation rates, in a given situation (or class of situations).

In this paper, rather than seeking a physical reason why aerosols affect precipitation in a particular way, we have instead investigated the structure that any mechanistic theory of an aerosol-cloud-precipitation interaction needs to have in order to describe an arbitrary change in surface rainfall statistics. We have shown that, despite the diversity in the possible precipitation responses, there is a fairly minimal set of statistical quantities which can describe any response. (At least, we hypothesise that this is the case, based on results from simulations with an aerosol-cloud microphysics scheme.) This set of quantities can be two moments of the rainfall rate distribution as functions of CDNC (a double-moment closure), or a single-moment and an inter-moment power law relationship that predicts a second moment (a single-moment closure). It is convenient to choose these moments to be the frequency of occurrence of rainfall (the zeroth moment) and the rainfall amount (the first

moment). Given these two quantities (or only rainfall frequency, in the single-moment case), the entire rainfall rate distribution can be reconstructed with an accuracy that is sufficient to resolve changes in the distribution due to aerosol concentration changes. This approach, using a small number of moments, is possible because there exists an invariant, i.e., 'universal', frequency density function for a normalised (non-dimensional) rainfall rate that is independent of CDNC and is unaffected by aerosol perturbations. The existence of this distribution, particularly its independence from background aerosol concentrations, significantly restricts the number of independent degrees-of-freedom that an aerosol-induced modification of rainfall can have. In particular, we have shown that if rainfall is partitioned into CDNC intervals, then an aerosol perturbation can affect the number of precipitating points and the mean-rainfall rate in each interval, but it cannot alter the probability distribution of rainfall fluctuations relative to the mean-rainfall rate. This is because the fluctuations are apparently governed by a universal distribution. Hence, two variables for each CDNC interval are sufficient to specify the rainfall distribution and its response to aerosol perturbations.

This analysis cannot predict whether or not precipitation increases or decreases in response to aerosol. For example, for our simulations we have not tried to explain why precipitation frequency increases with aerosol concentration for the deep-convective cloud regime. Rather, we have attempted to understand the relationship between changes in rainfall frequency and changes in other moments of the rainfall distribution. This relationship is fixed by the four aerosol-dependent constants needed to specify the frequency-intensity relationship of the regime, and by the universal distribution, $\Phi$. Any two systems in which these four constants were the same would respond to aerosol perturbations in structurally the same way, i.e., the dependencies of their rainfall distributions on rainfall frequency would be the same. Moreover, a theory that predicted these four constants, and predicted the response of rainfall frequency to aerosol, as a function of CDNC, would also predict the rainfall distribution change because this could be determined via the universal distribution.

Understanding how 'universal' the $\Phi$ distribution actually is, therefore constitutes a valuable question for future work. For the simulations used here the distribution is shown to be approximately independent of cloud-regime, e.g., it is the same for low-cloud only, and high-cloud dominated regimes. We have also shown that convection-permitting forecasts of subtropical and mid-latitude weather systems, and global climate simulations, all share the same non-dimensional rainfall distribution. The global climate simulations also show that the universal distribution is independent of whether rainfall is from the microphysics scheme or from the convection parametrization. A highly simplified model of rainfall as a multi-scale, stochastic process provides the tentative theoretical explanation that the universality is a result of geometric properties of rainfall time series. Perturbing an external factor (such as aerosols), modulates the properties of individual rainfall events, but the overall geometry of the rainfall time series changes in a statistically self-similar way. Understanding the extent to which this holds over a range of regimes, climate backgrounds, processes and time scales can be investigated further using observations and simulations with other models. We may expect that the invariant distribution is not independent of modeling system, and that it will break down for extreme parameter settings (e.g., no aerosols, or fixed-CDNCs), so a multi-model analysis and evaluation against observations would be a useful next step.

*Code and data availability.* The Python code and post-processed model data used are available for download from https://code.metoffice.gov.uk/trac/home

*Author contributions.* KF an PF jointly concieved the work and wrote the manuscript.

*Acknowledgements.* This work was supported by the UK-China Research and Innovation Partnership Fund through the Met Office Climate Science for Service Partnership (CSSP) China as part of the Newton Fund.

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

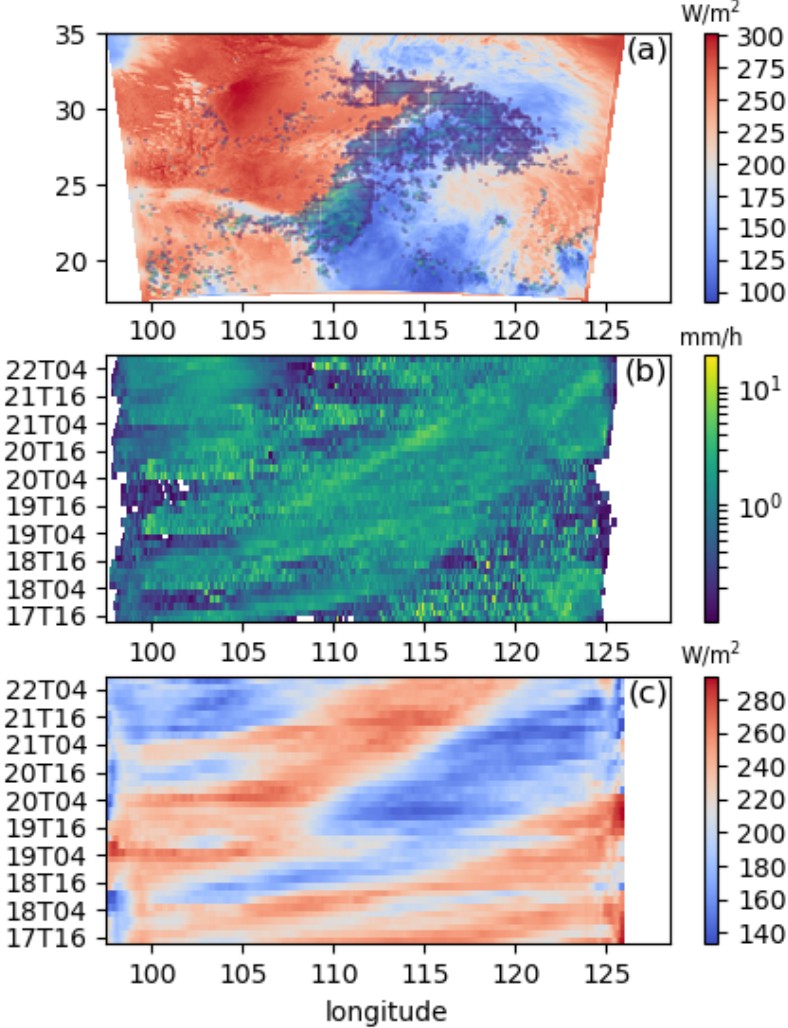

**Figure 1.** (a) The time-averaged multi-model mean outgoing flux of long-wave radiation (red-blue) and precipitation (green) at the top of the atmosphere between 00 UTC and 06 UTC on 20 May 2016. The averages are calculated from hourly instantaneous fields; precipitation rates less than 1 mm/h are excluded from the time averaging. (b,c) Hovmöller plots of merionally averaged surface rainfall rate, (b), and outgoing long-wave radiation, (c). Grid-points with precipitation rates less than 0.1 mm/h are excluded from the Hovmuller-plot mean shown in (b).

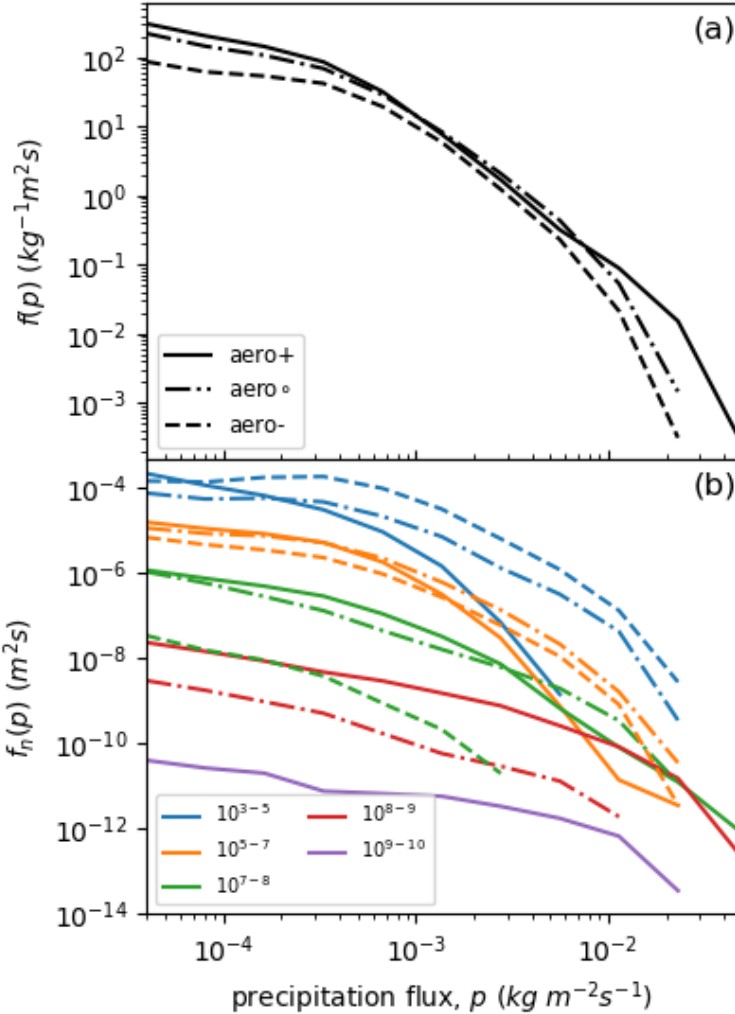

**Figure 2.** (a) The rainfall rate frequency distributions simulated by the high- (solid line), intermediate- (dot dashed) and low-aerosol concentration (dashed line) experiments in the deep-convective cloud regime and column-averaged CDNCs greater than $10^3 \mathrm{kg}^{-1}$. (b) The rainfall distributions decomposed into five CDNC intervals ($m^{-2}$; colors), the start and end-points of the intervals are shown in the panel's legend. For each interval (color), the line styles used in (a) indicate the CDNC-condition distributions for each model experiment (i.e., aerosol concentration).

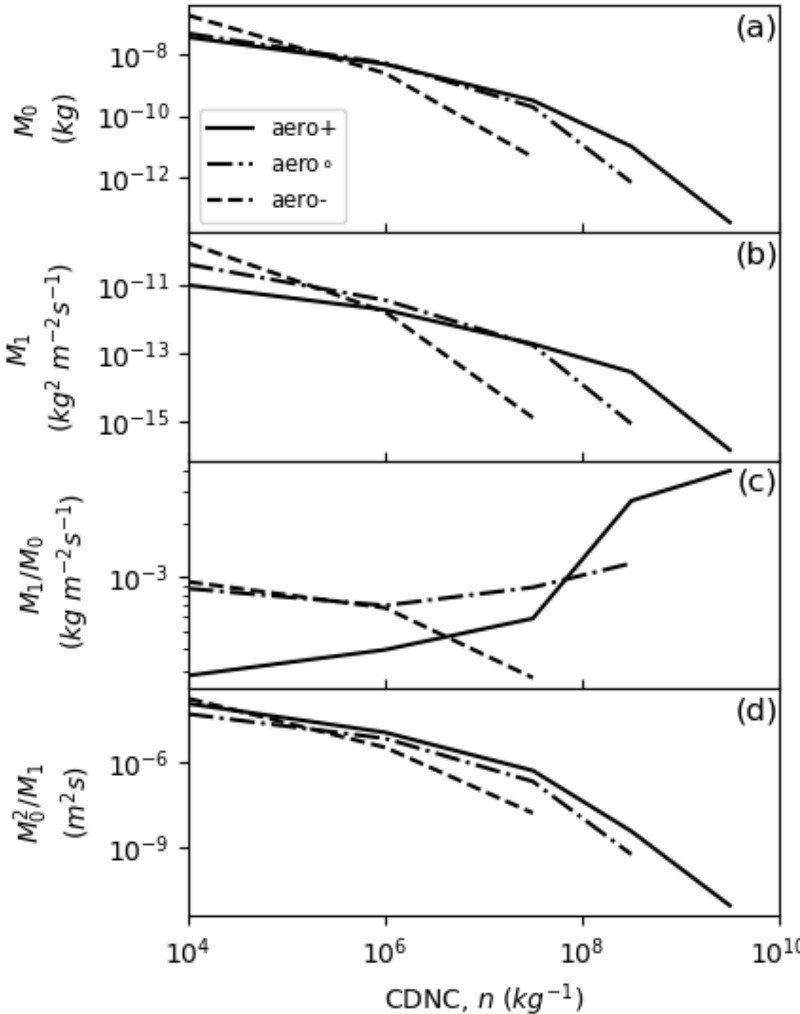

**Figure 3.** The moment-related properties of the CDNC-condition rainfall distributions for each model experiment, as functions of the column-averaged CDNC: (a) the rainfall frequency (per unit CDNC-interval), i.e., the zeroth moment, $M_0(n)$, of the CDNC-condition rainfall rate distribution, $f_n(p)$; (b) the CDNC-conditioned rainfall amount, $M_1$; (c) the CDNC-conditioned mean-rainfall rate, $\lambda_n = M_0/M_1$; (d) the normalised rainfall frequency, $\nu_n = M_0^2/M_0$.

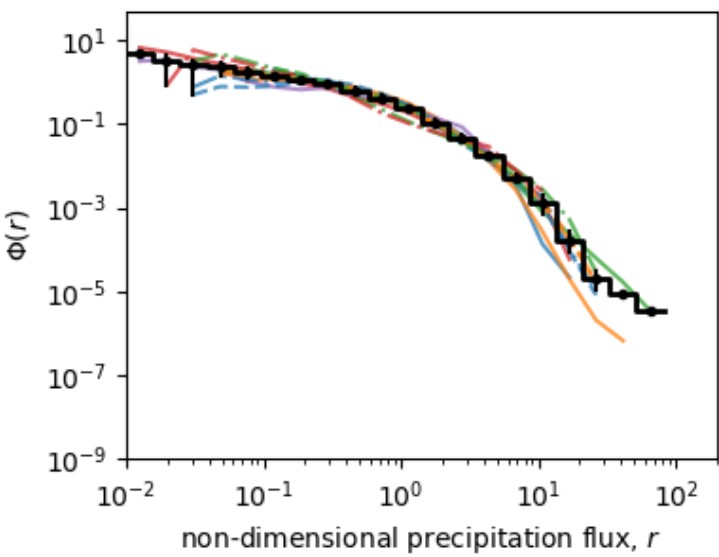

**Figure 4.** The re-scaled rainfall rate distributions for the five CDNC-concentration intervals (colors) and aerosol-concentrations (line styles) as functions of the normalised rainfall rate, plotted according to the conventions established in Fig. 2, and the average histogram (black line). The vertical black bars in each bin of the histrogram show the inter-model spread, i.e., the range of values obtained by defining an average histogram for each model experiment seperately (and hence the sensitivity of the dimensionless distribution to aerosol concentration).

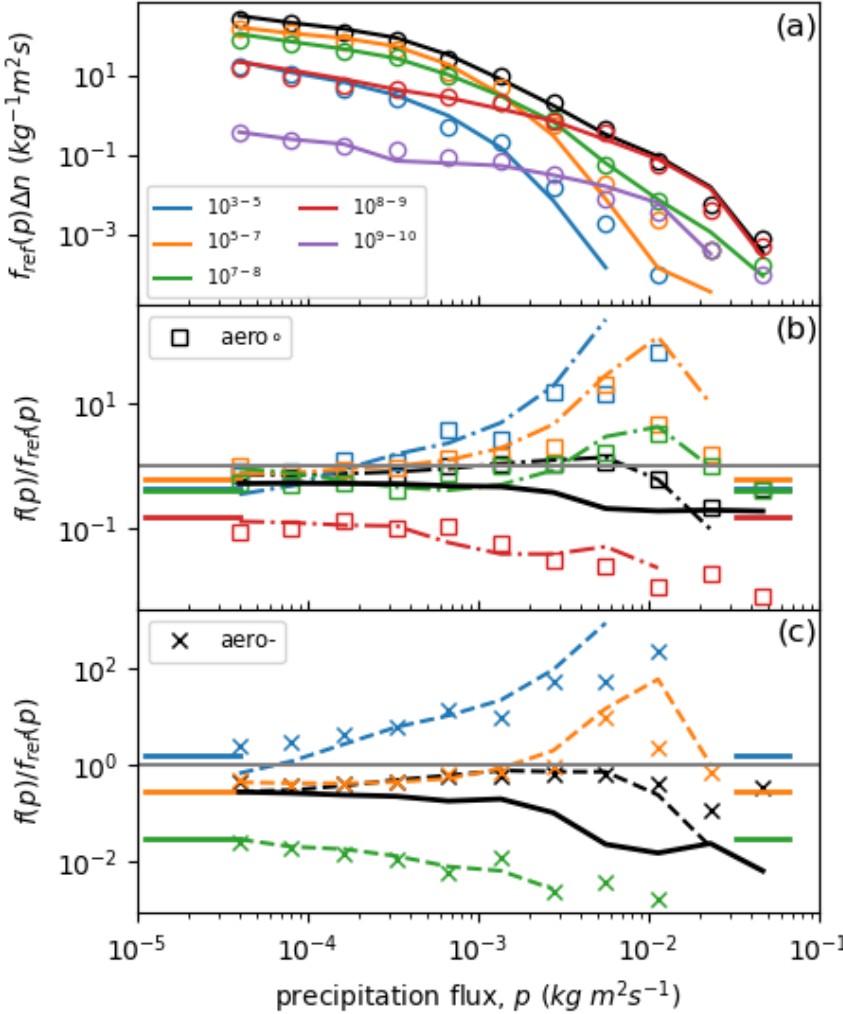

**Figure 5.** The double-moment reconstructions of the rainfall rate distributions in the deep-convective cloud regime. (a) The simulated (lines) and reconstructed (circles) rainfall distributions for the high-aerosol concentration experiment. The colored lines show the CDNC-condition contributions to the total rainfall distribution from the CDNC-intervals in the panel's legend. The black line is the total rainfall frequency (the sum of the colored lines). (b,c) The simulated and reconstructed differences between the rainfall distributions in the intermediate- and low-aerosol concentration experiments and the high-concentration experiment. The solid black lines show the reconstructed differences obtained if the CDNC-conditioned mean rainfall rates, $\lambda_n$, are assumed to be unchanged by the aerosol perturbations. The short colored bars indicate the changes in the normalised frequencies, for each CDNC interval, relative to the high-aerosol experiment.

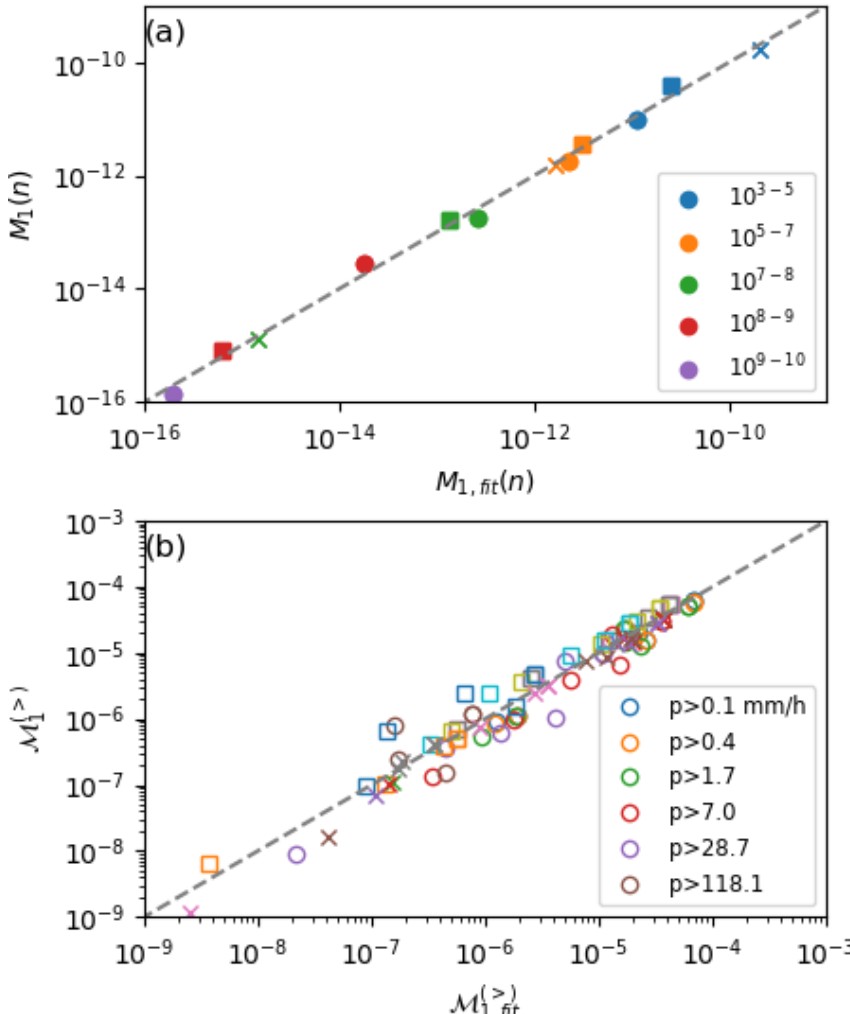

**Figure 6.** The single-moment reconstructions of the rainfall rate distributions. (a) The simulated rainfall amounts (vertical axis), compared to the values predicted by the single-moment closure, using the aerosol-concentration dependent power-law relationships between the zeroth and first moments of the CDNC-conditioned distributions. The circles, squares and crosses correspond to the high-, intermediate and low-aerosol concentration expriments, respectively. The colors indicate the CDNCs. (b) The simulated (vertical axis) upper-partial first moments of the rainfall rate distributions, compared to the values predicted by the single-moment closure. The colors indicate the precipitation-rate thresholds above which rainfall is accumulated.

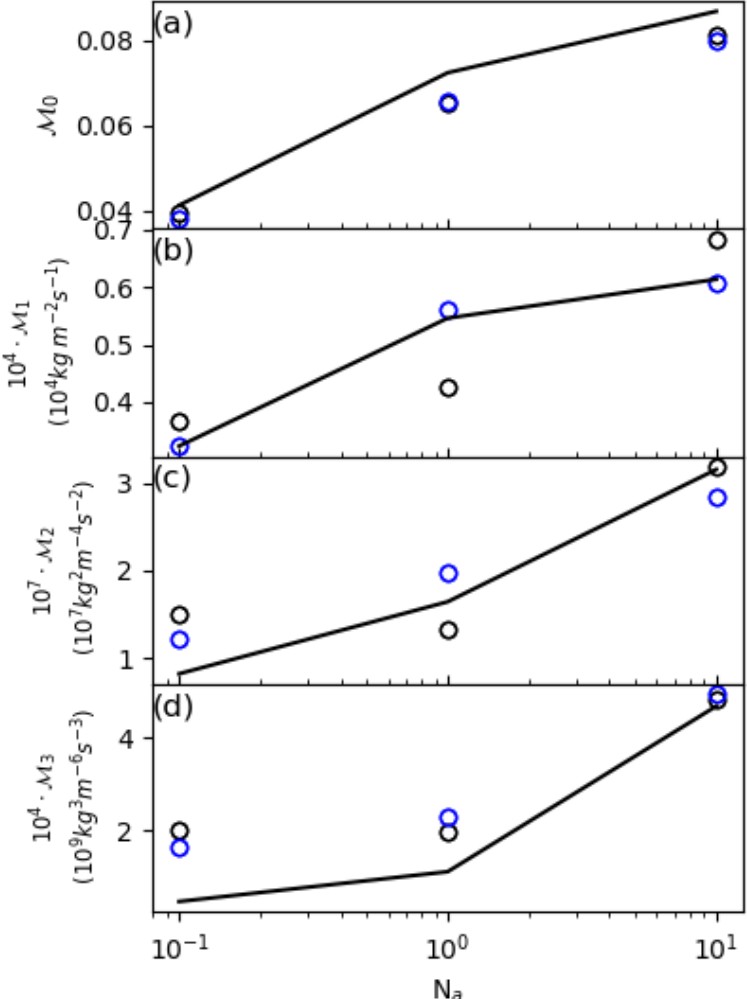

**Figure 7.** The simulated (lines) and predicted (circles) bulk-rainfall characteristics as functions of the (normalized) aerosol-concentrations, $N_a$, for the three model experiments: (a) mean rainfall frequency; (b) the mean rainfall amount, scaled by $10^4$ for presentational convenience; (c,d) the second- and third-moments of the rainfall frequency distributions. The blue and black circles are double-moment and single-moment predictions, respectively.

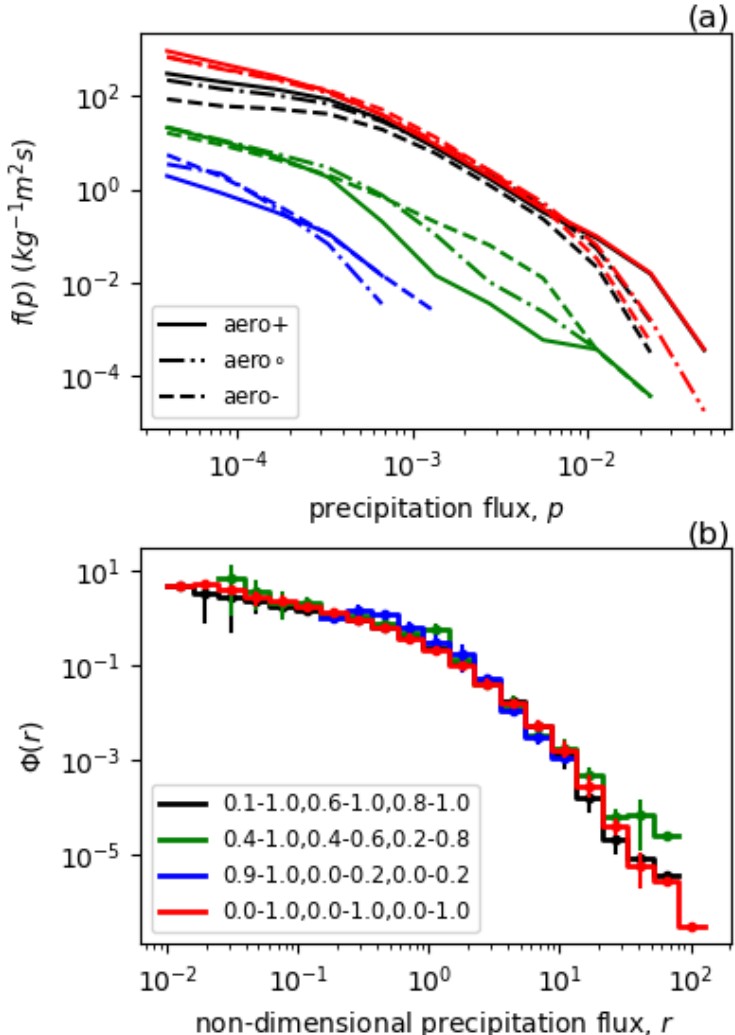

**Figure 8.** The regime dependence of the rainfall rate distributions for the deep-convective (black lines), low-cloud dominated (blue lines), and intermediate (green lines) cloud regimes. The red lines show the distributions for the whole domain (all cloud-fraction combinations). (a) The simulated rainfall rate frequency distributions for the three regimes of fractional cloudiness and the total (whole domain) distributions. (b) The histograms of the universal distributions for each regime, with the inter-model ranges indicated by the vertical bars in each non-dimensional rainfall rate interval.

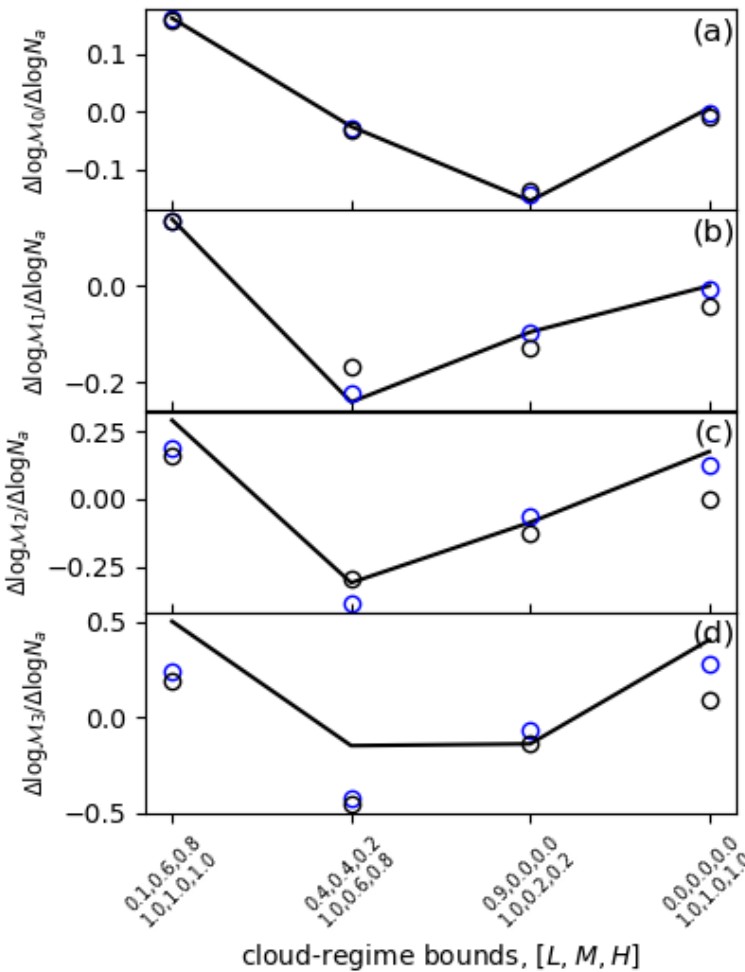

**Figure 9.** The sensitivities of bulk rainfall characteristics to aerosol perturbations for each cloud-fraction regime (horizontal axis). (a) The sensitivity of mean rainfall frequency; (b) the sensitivity of mean rainfall amount; (c,d) the sensitivities of the second- and third-moments of the rainfall rate distribution. The blue and black circles show the sensitivities predicted by the double- and single-moment closures, respectively.

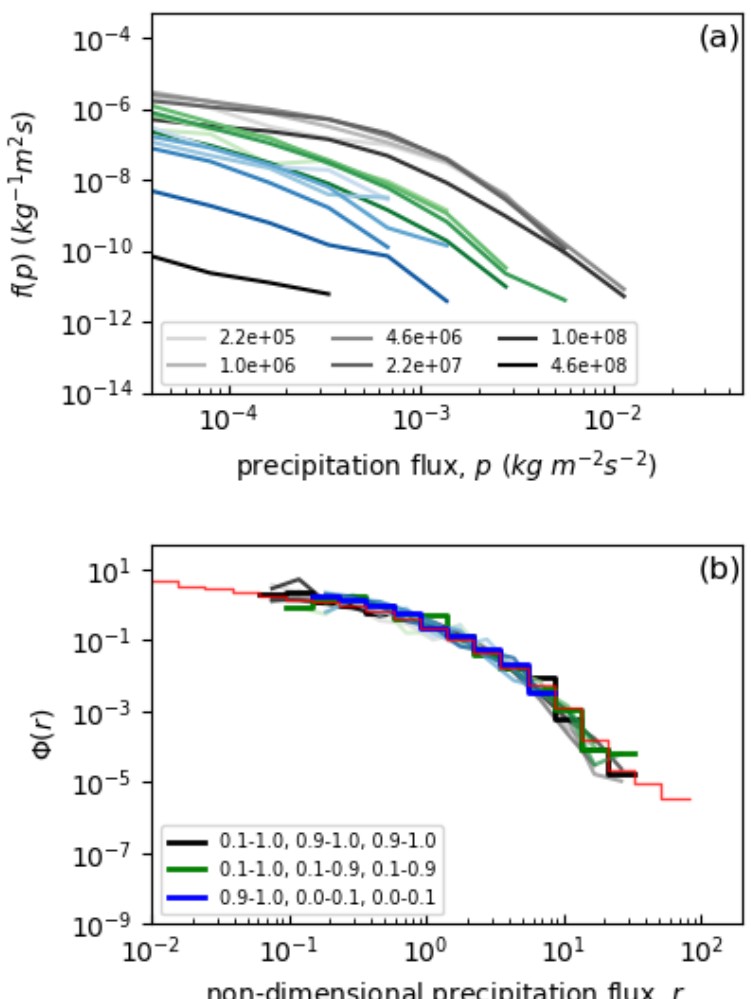

**Figure 10.** The CDNC dependencies of the rainfall rate distributions in the twenty-three, UK forecasts for three mid-latitude cloud regimes (deep-frontal (black lines), low-cloud dominated (blue lines), and intermediate (green lines)). (a) The simulated rainfall rate frequency distributions, for each CDNC interval, for the three regimes of fractional cloudiness. The degree of shading for each color show the CDNCs. (b) The rescaled distributions (lines) and average universal histograms for each regime. The red line shows the non-dimensional distribution from the May 2016, subtropical case study.

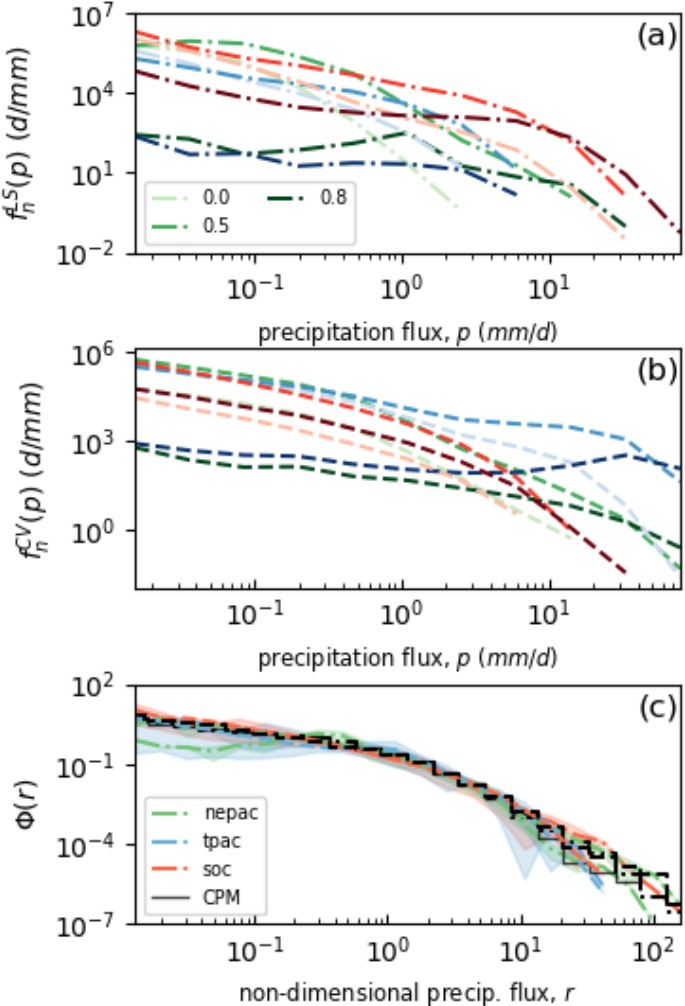

**Figure 11.** The cloud-albedo dependencies of the rainfall rate distributions in a 20 year AMIP simulation for three geographical regions (the north-eastern Pacific (green lines), the tropical Pacific (blue lines), and Southern Ocean (green lines)). The simulated large-scale rainfall, (a), and convection-scheme rainfall, (b), frequency distributions for each cloud-albedo interval, for the three regions. The degree of shading for each color show the CDNCs. (c) The rescaled distributions (lines) and average universal histograms (black lines) for each regime. The solid black line shows the non-dimensional distribution from the May 2016, subtropical case study.

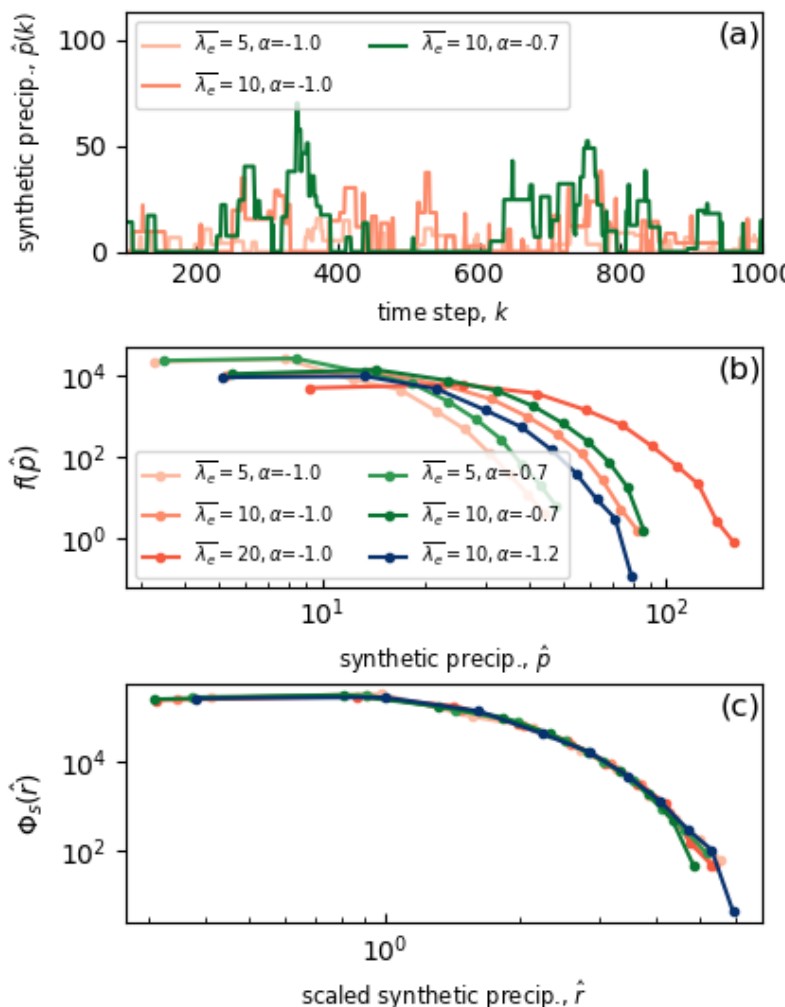

**Figure 12.** The parameter dependencies of the synthetic rainfall rates generated by the stochastic rainfall generator for three values of the duration-exponent parameter, $\alpha$ (colors), and three values of the event-mean intensity, $\lambda_e$ (shading). (a) Sub-sections of the time series of synthetic rainfall for three of the parameter combinations. (b) The frequency distributions of synthetic rainfall. (c) The renormalised distributions of the scaled synthetic rainfall.

**Table 1.** The parameters needed for fitting $M_2$ and a function of $M_1$ and AC for each cloud regime

| $LMH$ min | $LMH$ max | $\epsilon$ | $\eta$ | $\gamma$ | $\delta$ |
|---|---|---|---|---|---|
| 0.1 0.6 0.8 | 1.0 1.0 1.0 | -3.57 | -1.46 | -0.164 | 0.957 |
| 0.4 0.4 0.2 | 1.0 0.6 0.8 | -2.51 | -1.65 | -0.157 | 1.1 |
| 0.9 0.0 0.0 | 1.0 0.2 0.2 | -4.61 | 0.5 | 0.0457 | 0.934 |
| 0.0 0.0 0.0 | 1.0 1.0 1.0 | -3.82 | -1.41 | -0.163 | 0.942 |