# Peer review of "A strong statistical link between aerosol indirect effects and the self-similarity of rainfall distributions"

_Atmospheric Chemistry and Physics, 2021_

## Author Response (AR1)

**Replies to Review 2**

This study describes what the authors term a "universal" self-similar probability density function of re-scaled rainfall rate/intensity and its invariance under aerosol perturbations. It is an intriguing idea that seems to be well-supported by the analysis of a single large-domain simulation. While I agree with the authors that it is beyond the scope of this work to simulate a multitude of cases with a variety of models, etc., I find the assertion of "universal" scaling from one simulation to be quite a stretch. Either the language needs to be changed, or some other evidence given that the "universal function" will hold for different storm types/climatological contexts.

I agree with this conclusion; we've tried to strengthen the claim of 'universality' by adding a new section (Section 7.3) which includes the following:
1) the universal distribution for 20 case studies of mid-latitude weather, using a convection-permitting (1.5km resolution) regional model over the UK;
2) universal pdfs for three large (17 x 12 degree) regions, covering tropical Pacific, north-eastern Pacific, and Southern Ocean, from 20-years of daily mean precipitation from AMIP simulations with a global climate model.

The inclusion of the UK cases samples a variety of different meteorological regimes. The global simulation samples a different microphysics scheme (a single-moment scheme), and convection-scheme rainfall, in three different climatological backgrounds (tropical deep convection, subtropical stratocumulus, and mid-latitude storm tracks). The results show that the 'universal' distribution is approximately independent of the factors sampled.

Another obvious perturbation that would increase confidence in the universality assertion would be to perturb sensitive parameters of the microphysics scheme (akin to what the authors did in Furtado et al. 2018).

The comparison to the UK NWP cases encompasses a different aerosol microphysics scheme (but the same cloud microphysics). The global AMIP simulations use a different cloud microphysics (the single moment scheme used in Furtado 2018) and a convection parametrization. So, the similarity of the non-dimensional pdf across these simulations strengthens the climate that

some form of 'universality' is at work.

As it is, I believe the title puts it best: there is evidence for a strong statistical link between AIE and self-similar distributions, but I am not convinced that this is the last word on the characteristics of the underlying distribution.

**I agree with this; I've added a few sentences to the last paragraph of conclusions to this effect.**

I recommend the study for publication pending the authors' response to the above critique and several minor and typographical comments below.

Minor/typographical comments All corrected L118: "were performed" instead of "where performed" L182: "suppression becomes stronger" L199: "referred to as rainfall intensity" L200: "where rain is falling" L201: "CDNC-conditioned mean rainfall rate"

L208: "up to four orders"

L252: "the sum of CDNC-conditioned"

L270: "assumption corresponds to the simplification"

L292: "fewer than two moments"

L301: I am confused – the parameters in Table 1 have different symbols. Please correct.

Thanks! These are parameters needed to get the prefactor *x* and exponent *y* parametrically from the aerosol number, *Nsol*; we've corrected the description in the text (Eq. 11) and symbols used in the Table.

L301: Do you mean Figure 6b? Hard to tell because it looks like the axis labels are wrong. **I've corrected the axis labels in 6a.**

L305: There is no factor of M2(n) in Eq. 7. Do you mean M1(n)?

**Yes, M1(n); corrected**

L310: I think you mean "M0,...,M3"

L319-321: What is your metric for "capturing the trends" in Fig. 7? It looks to me like you get a great fit for M0 and then the fit degrades with increasing moment order. Even M1 is pretty far off for single moment Na=1.

**Agreed. We've revised the discussion of this Figure:** "In most cases, the predictions are able to reproduce the simulated values of the moments reasonably well. The agreement is slightly less good for some values of the single-moment reconstructions and for the highest-order moment tested."

L351-353: Scale breaks are common in systems like these due to both statistical noise and violations of scaling laws. Can you rule out the latter?

**We can't rule this out – it is possible; I think this possibility is covered by our concluding remarks about "understanding the extent to this [universality] holds"**

L391: "the probability distribution of"...of what? corrected

L396: "We do not if the distribution is..."

L404: Should there be an "and" between the definitions of r1 and r2? Yes! corrected

L414-415: This sentence is confusing. I suggest you break it into two and reword.

**Simplified to: "The choice of M\_0 and M\_1 is arbitrary: as shown by Field and Shutts (2009),**

any pair of moments could be used for the reconstructions."

L416: "a family of power-law relationships" corrected

L424: "a detailed understanding of how aerosols..." corrected

L427: "rather than seeking a physical reason for why aerosols..." corrected

L434: "choose these moments" **corrected**

L460: "next" instead of "nest" corrected!

Fig. 4 caption: "and hence the sensitivity of..." corrected

Fig. 6a: should axes read "M1, fit" and "M1" instead of referencing M2? corrected

Fig. 9: I am confused about which regime is which in the figure. Could you descriptively label the x-ticks instead of the visually-distracting cloud fraction bounds?

**I hope it's ok, I've left the quantitively descriptive labels; I know they are a bit cumbersome, but I think they help with reproducibility of the results.**

Citation: https://doi.org/10.5194/acp-2021-443-RC2

**Replies to Reviewer 1**

The manuscript describes the derivation of an invariant distribution of surface rainfall rate as a function of the precipitation flux. The derivation is based on a set of numerical model simulations where the aerosol number concentrations are varied. The resulting invariant distribution appears to be independent of the amount of aerosol and has the ability to predict the response of the rainfall statistics to a perturbation of the aerosol. The manuscript describes an interesting work to construct such an invariant and fits into the scope of ACP. I recommend the manuscript for publication after the authors have addressed the following comments and revised the manuscript accordingly.

**General comments:**

In the manuscript, you state at multiple locations that you construct a universal distribution for the rainfall statistics. However, the whole analysis is based on three simulations of one case. At the very end (starting at line 455) you mention some of these serious assumptions that might limit the conclusion to only this single simulated case. At the moment, your results show that for your model and for this simulated case there is a universal distribution. To get an idea about the true universality of the result, one should

- simulate more cases with your model

- in particular also at different locations in the world (maybe the universal distribution is different in the tropics or in the arctic?)

**- ideally using also different models.**

**To get an idea about the universality of the result, we've added a new section (Section 7.3) that:**

 constructs the universal distribution for 20 case studies of mid-latitude weather, using a convection-permitting (1.5km resolution) regional model over the UK;
 constructs universal pdfs for three large (17 x 12 degree) regions, covering tropical Pacific, north-eastern Pacific, and Southern Ocean, from 20-years of daily mean precipitation from AMIP simulations with a global climate model.

The inclusion of the UK cases samples a variety of different meteorological regimes. The global simulation samples a different microphysics scheme (a single-moment scheme), and convection-scheme rainfall, in three different climatological backgrounds (tropical deep convection, subtropical stratocumulus, and mid-latitude storm tracks). The results show that the 'universal' distribution is approximately independent of the factors sampled.

I know that this is maybe far too much work, however I think at least a second case should be simulated and the predictions made with your present universal distribution be compared to the actual rainfall statistics in the second scenario. But even then, the rainfall statistics might be much more dependent on the implemented microphysics scheme.

Reading through your data analysis, I wondered how robust the statistics are, i.e. are there enough clouds to sample from in each of the regimes you mentioned?

The extend analysis of two other model configurations, different regions and more cases, supports the conclusion that the existence of a widely applicable re-scaling is statistically robust.

We've also added a new section (Section 7.4) which gives a theoretical argument for the existence of universal scaling, based on a stochastic rainfall generator. The generator treats rainfall at each point as a sum of independent precipitation 'events' with durations and intensities drawn at random from underlying power-law probability distributions. We show that the stochastic model also has data-collapse to a universal distribution.

Specific comments:

Line 25 to 28: There are also studies that indicate that, on average, the amount of precipitation is not influenced due to the buffering effect of clouds. The idea of buffering is described in Stevens and Feingold (2009) and a study indicating that there might, on average, be no influence is Seifert et al (2012) (although the latter study is based on an operational NWP model, i.e. the coupling to aerosol is limited).

We've added the following text to the Introduction: "Furthermore, Stevens and Feingold (2009) suggested that in some situations the answer to this question [of whether ACIs increase/decrease precipitation] is 'neither', because systems of clouds adjust to counteract the aerosol-induced changes in precipitation. This implies that although an individual cloud may have a large response to aerosols, changes in the amount of precipitation average over an area may be much smaller. This was illustrated by Seifert et al (2012) who showed that aerosols had negligible effects on precipitation over a large range of region numerical weather predictions."

Hence at this point it is important to state the context more clearly: Do you ask the question for a specific cloud? Do you ask the question for the amount of precipitation averaged over an area?

Added explicitly after the bullet points of "aims" in the Introduction: "We will primarily consider these questions in relation to the area- and time-averaged statistics of rainfall over a large domain for a case of typical case of monsoon rainfall over East Asia."

See also the paragraph starting at line 90. *I've added a citation to Siefert et al (2012) in this paragraph.*

Line 30 to 32: I do have problems understanding this sentence. Not every cloud produces rain and a cloud is an example of a system with unbalanced sources and sinks (otherwise the cloud would not have formed)? Please clarify.

**I think there is an implicit steady state assumption in Khain's 2008 framework which implies balanced (i.e., steady state) systems where sources and sinks are approximately equal over a period; it therefore probably doesn't apply well to the formation/dissipation of individual clouds. I've tried to clarify this as follows:**

"If the precipitation rate adjusts over time to a slowly varying state in which sources and sinks of condensate approximately cancel out, then an aerosol change which increases the sources more than it increases the sinks will necessarily lead to an increase in the amount of precipitation. Therefore, if we consider two systems both precipitating at rate, P, and subject one to an aerosol perturbation, the perturbed system will evolve to a new state with a (possibly) different precipitation rate, P + dP, where dP is due to a change in the net source of condensate relative to the unperturbed system." Line 120 to 121: You refer to the domain of your simulation by pointing the reader to a plot of radiative fluxes. You should either only state the domain of your simulation by indicating the geographical coordinates or adding a geographical map. I prefer the latter.

**I hope it okay that I've opted to specify the coordinates in the text ( $17 \circ -35 \circ N,97 \circ -126 \circ E$ ) because there are already quite a few figures.**

Line 127 to 128: To which degree do your results degree on the choice of the lateral boundary conditions for the aerosols?

**Results for a regional NWP over the UK (Section 7.3) which uses a different aerosol model, suggest that scaling of the distributions is independent of LBC details.**

Line 296: You refer the reader to a figure in the supplementary material; please include the figure in the main text.

**I'd like to keep this in the supplementary, if possible (there are so many figures already)**

Line 317: The predictions do not always reproduce the simulated values, e.g. the black circle in panel b is off; also in panel d a more stagnant behaviour is predicted instead of the decrease that is visible in the solid line.

**Agreed, the original was too strong – I've revised it to:**

"In most cases, the predictions are able to reproduce the simulated values of the moments reasonably well. The agreement is slightly less good for some values of the singlemoment reconstructions and for the highest-order moment tested."

Caption of Figure 4, fourth line: The sentence within the brackets appears exaggerated to me. You can only assess the sensitivity to your experiments, which is different from a "universal" sensitivity. *Change "universal" to "dimensionless"*

Figure 6: Is there a motivation for the thresholds used? Why not use "simpler" values, e.g. 0.5 instead of 0.4 or 29 instead of 28.7?

**I choose cloud regime boundaries to isolate the rainfall regimes shown in S3 as well as possible (See discussion on lines 357–360)**

**Technical corrections:**

Thank you very much for finding all these! All corrected (except for Fig. 9 for which I'd rather keep the full labels, if possible, because they are more descriptive)

- Line 36: "it is" should read "is"
- Line 86: interpreted
- Line 118: "were" instead of "where"
- Line 175: based on
- Line 182: becomes
- Line 221: "colored lines in figure 4"
- Lines 248 and 288: The section numbering should read 5.1 and 5.2 instead of 5.0.1 and 5.0.2
- Line 267: There is a period missing after the equation.

- Line 292: In particular
- Line 310: It should read M\_0, ..., M\_3
- Line 323: fractions
- Line 331: Delete one of the "because"
- Line 343/344: It should read "...precipitating (and highly cloudy) regime, ..."
- Line 356: that the universal
- Line 363: It should read \gamma\_0 \gamma\_3 instead of \gamma\_{0-3}
- Line 366: Index k should only range between 0 and 3.
- Line 387: "...parameter space is a..."
- Line 392: Delete "of"
- Line 396: "if the distribution"
- Line 460: next step
- Figures 1, S1, S2, : There are missing labels for the axes.
- Caption of Figure 2, second line: column-averaged
- Figure 2b: Units are missing in the legend. I suggest to add the units in the caption.
- Caption of Figure 4, fourth line: "sensitivity" instead of "sensitive"

- Figure 9: I suggest to also indicate the regime in the axes label instead of only the values for L, M, H.

- Caption of Figure 9, third line: distribution
- Caption of Figure S4, first line: M\_1 should read M\_0?

**References:**

Seifert, A., Köhler, C., and Beheng, K. D.: Aerosol-cloud-precipitation effects over Germany as simulated by a convective-scale numerical weather prediction model, Atmos. Chem. Phys., 12, 709–725, https://doi.org/10.5194/acp-12-709-2012, 2012. Stevens, B., Feingold, G. Untangling aerosol effects on clouds and precipitation in a buffered system. Nature 461, 607–613 (2009). https://doi.org/10.1038/nature08281 Citation: https://doi.org/10.5194/acp-2021-443-RC1